# Household Dietary Diversity among Households with and without Children with Disabilities in Three Low-Income Communities in Lusaka, Zambia

**DOI:** 10.3390/ijerph20032343

**Published:** 2023-01-28

**Authors:** Mary O. Hearst, Leah Wells, Lauren Hughey, Zeina Makhoul

**Affiliations:** 1School of Nursing, University of Minnesota, Minneapolis, MN 55455, USA; 2Public Health Department, St. Catherine University, St. Paul, MN 55105, USA; 3SPOON Foundation, Portland, OR 97214, USA

**Keywords:** household dietary diversity score, children with disabilities, Zambia

## Abstract

The purpose of this manuscript is to describe household dietary diversity (HDDS) in Lusaka, Zambia between households with and without a child with a disability living in the same communities. Cross-sectional data were collected in three low-income compounds in September 2021. Participants included households with a child with a disability enrolled in Kusamala+, a community-based program, (*n* = 444) and a convenience sample of adults living in the same area without a child with a disability (*n* = 1027). The HDDS tool asked about food groups consumed in the past 24 h by people in the household. The responses were summed (yes = 1, no = 0), range 0–12. Individual dietary diversity scores (IDDSs) were calculated for children (0–8 items). Analysis included descriptive statistics and linear regression. Mean HDDS for the households with a child with a disability was 4.8 (SD 2.1) vs. 6.1 (SD = 2.2) among households without a child with a disability (*p* < 0.001). The individual score for children (IDDS) for households with children with disabilities was 2.6 (SD = 1.4) vs. 3.7 (SD = 1.6) for households without a child with a disability. Households with a child with a disability had a significantly lower HDDS and IDDS in unadjusted and adjusted models (*p* < 0.001). National policy must assure the most vulnerable populations, and often hidden, receive focused financial and food support.

## 1. Introduction

The COVID-19 pandemic exacerbated food and nutrition insecurity for many households already struggling with dietary diversity. Households lacking consistent access to food experience reduced quality and diversity in their diet, which is a critical determinant of childhood nutritional status [1,2]. A report from the UN Food Systems Summit estimates that less than one in four young children in Eastern and Southern Africa are being fed a minimally diverse diet [1]. In Zambia, located in sub-Saharan Africa, severe poverty among a significant population of Zambian households creates barriers to achieving sustainable food security and consuming a diverse diet [3,4]. Among households with a child with a disability, the situation is more complex.

Households with a child with a disability face unique challenges in assuring dietary diversity in the household. For example, a primary caregiver may be reluctant or unable to leave the child to find work or gather food, limiting food availability in the household [5]. It is also known that children with disabilities are more likely to have nutritional deficiencies [6,7] particularly with conditions that impact the ability to chew and swallow foods, such as cerebral palsy or cleft palate [8,9]. Discrimination may result in the child with a disability being fed smaller quantities or less nutritious foods than other siblings believed to have greater potential for survival and contribution to the household [8]. Consumption of diets that are not diversified have been linked to high stunting and wasting in children [10], and reduced optimal cognitive development [11]. Alternatively, more diversified diets may also be higher in energy, especially in low-income countries and communities, thus being more obesogenic [1,12]. The household dietary diversity score is a proxy measure of household economic ability to access a variety of foods and is an indicator of food security in general [13].

Pre-existing and co-occurring vulnerabilities further exposed families of children with disabilities in the context of the COVID-19 pandemic [14]. The first COVID-19 cases in Zambia were recorded in March 2020 [15] when farmers were still recovering from the 2018–2019 drought and flash floods that disrupted food supply chains, affecting 2.3 million people [16]. Already vulnerable economic, healthcare, transportation, and education systems were aggravated by the pandemic and its subsequent mitigation measures [16]. Loss of income due to movement restrictions, partial closure of informal markets, and increased food prices left families with limited access to food sources [14]. The prevalence of acute food insecurity increased between July–September 2021 to approximately 1.2 million [3] people living in Zambia, with most being affected in Luapula, Lusaka, and western provinces. Data for this study were collected September 2021. The timing and context of data collection is important to distinguish as a potentially exacerbating factor.

The degree of the impact of COVID-19 on the ability to access food and household dietary diversity among households with a child with a disability, and households without, is unknown. Yet, dietary diversity cannot be discussed at this time without the context of COVID-19. The purpose of this paper is to assess the household dietary diversity among low-income families, with and without a child with a disability, living in three low-resource compounds (one is divided into two regions) in Lusaka, Zambia. See Figure 1. Data were collected as part of an ongoing community intervention and at the time of acute food insecurity [3]. The hypothesis is that families with a child with a disability have lower levels of dietary diversity, compared to families living in the same economic and environmental context without a child with a disability. This manuscript has important implications for data-driven community-based policies to enhance the life chances of the most vulnerable.

## 2. Materials and Methods

The data presented are from endline data collection following a two-year program called Kusamala+ (“to care” in the local language of Nyanja), a community-based program by the Catholic Medical Mission Board (CMMB) Zambia and St. Catherine’s University (St. Paul, MN, USA). Kusamala+ is currently active in Kanyama, Misisi, and Chawama—three low-income and low-resource communities in Lusaka. Kusamala+ increases local capacity to decrease stigma toward children with disability and their families, strengthens healthcare systems around disability; and builds capacity for home-based care, community building, and referral systems [17,18]. Referrals were to the health facility, physiotherapy, Zambian Association for Persons with Disability, and Ministry of Social Welfare, the latter for opportunity for cash transfer. HDDS was added to the endline data collection because a rapid health assessment completed in 2021 indicated COVID-19 impacted access to food and future investigation was needed [14]. There was no food or cash distribution.

### 2.1. Participants

*Households with children with disabilities.* Households currently active in the program (*n* = 436) between September and October 2021 were contacted via the Kusamala+ community healthcare workers (community caregivers: CCG) and CMMB trained data collection staff to complete an endline survey evaluating the impact of the Kusamala+ program on family quality of life and community-based stigma. The primary caregivers were contacted by phone or direct home visit complying with local COVID-19 precautions. If the caregiver or other adult was not at home, a second visit was attempted. *Households without a child with a disability.* Following each interview with the household with a child with a disability representative, CMMB staff approached adults living in households without a child with a disability, using a convenience sampling approach from people walking by or at nearby housing. In the compounds, there is heavy foot traffic along with people sitting outside their homes engaged in daily life tasks, such as laundry, food preparation, and other chores. The compounds are very low-income, 87% share a latrine, half of the residents collect water from a community tap, and there are limited regular employment opportunities. The two populations were drawn from a similar socioeconomic context although differences may exist at the household level [19,20].

### 2.2. Measures

The HDDS was released in 2008 as a population-level indicator of household food access [13,21]. Household dietary diversity is an indicator of food access and utilization of food to meet the nutritional needs of the household for a productive life [13]. Household dietary diversity is associated with healthier birth weight [22], child growth indicators [23,24] and higher hemoglobin concentrations [13,21,25,26], although other factors also contribute to the presence of anemia [27]. Dietary diversity is also highly correlated with total caloric intake and protein adequacy [13,21].

The HDDS instrument instructions instructs the data collector to ask the person responsible for food preparation, or another adult who was present and ate in the household the previous day, about all the foods eaten inside the home during the previous day and night, by any member of the household. Particularly in lower income households, dietary intake relies primarily on staple foods and is consistent over time [28]. The twelve categories include (1) cereals, (2) roots and tubers, (3) vegetables, (4) fruits, (5) meat, poultry, offal, (6) eggs, (7) fish and seafood, (8) pulses, legumes, nuts, (9) milk and milk products, (10) oil/fats, (11) sugar/honey and (12) condiments. The instructions for the cereals is as follows: “Any [INSERT ANY LOCAL FOODS, E.G., UGALI, NSHIMA], bread, rice noodles, biscuits, or any other foods made from millet, sorghum, maize, rice, wheat, or [INSERT ANY OTHER LOCALLY AVAILABLE GRAIN]? This food group thus allows for local food sources to be named.

Each food group is scored as a “1” for “consumed” and “0” for “not consumed”. The score is summed. Average HDDS is the sum of individual households’ scores divided by the total number of households surveyed. The individual dietary diversity score (IDDS) is used as a proxy measure of nutritional quality of an individual’s diet and is commonly used to measure the nutritional quality of children’s diets [13,21]. The IDDS food groups are combined with a score of 1 = consumed and 0 = not consumed for the following groupings: (1) grains, roots or tubers; (2) meat and fish; (3) fruit; (4) vegetables; (5) eggs; (6) beans; (7) dairy; and (8) fat. Sugar, honey, and condiments, coffee and tea are excluded because they do not significantly contribute to the child’s diet quality [29]. The ranges for HDDS and IDDS are 0–12 and 0–8, respectively, with higher numbers indicating greater diversity. Target levels of diversity are not available. Instead, an HDDS target can be calculated by creating HDDS tertiles, calculating the average of the highest tertile, and using that value as the ideal or target HDDS values. The same process was carried out for the IDDS using the average scores of the 33% of households with the highest diversity [26].

Data were collected using REDCap mobile app [27,28]. An adult from the household with a child with a disability was asked to complete the survey, provided verbal consent upon agreement and the staff member read the questions aloud. The survey took 20–30 min to complete and was conducted in Nyanja, Bemba or English. Households with a child with a disability were asked demographic information, questions about the child’s disability and function, community support, stigma, quality of life [29,30,31,32], the extent to which COVID-19 impacted aspects of daily life including access to food, feeding and nutritional challenges, and the household dietary diversity score (HDDS) [14]. Households without a child with a disability were asked questions related to stigma and HDDS. The staff member approached adults in the community at random and explained the purpose of the survey and if interested, the individual verbally consented to participate in the 10-min survey read aloud by the CMMB staff conducted in Nyanja, Bemba or English.

Given the context of the COVID-19 pandemic, all participants were asked to what extent COVID-19 impacted their ability to access food with response options of (1) to a great extent, (2) to a small extent, (3) it has not changed, and (4) it has improved. Households with children with disabilities were asked if their child had difficulty with feeding, which contributes to nutritional deficits and reduced dietary diversity. All participants were asked their household source of water and if they had soap in the house. Additional questions included documenting the compound, sex (binary) and age for both household types. St. Catherine University and the University of Zambia IRBs approved these protocols.

### 2.3. Analysis

Data were exported from REDCap into Stata v15.1 (StataCorp, College Station, TX, USA) and appended into one file for analysis. Demographic descriptive data and impact of COVID-19 were summarized and compared using *t*-tests and chi-square statistics with a significance level of *p* ≤ 0.05 indicating a statistical difference. Water source and presence of soap in the home was compared between household types as a marker of economics. HDDS and IDDS scores were summed per description in the methods. Target values were calculated by creating HDDS tertiles, calculating the average of the highest tertile of dietary diversity, and using that value as the ideal or target HDDS values. The same process was carried out for the IDDS [26]. Unadjusted and adjusted linear regressions compared the HDDS and IDDS scores by participant with or without a child with a disability and adjusted for sex, age and impact of COVID-19.

## 3. Results

A total of 436 housholds with a child with a disability who were enrolled in the program and 1027 households without a child with a disability consented to and completed the survey. The average age of the respondent from the household with a child with a disability was significantly older than the respondent from a household without a child with a disability (39.3 years vs. 37.9 years old). The vast majority of respondents from the households with a child with a disability were female (95.3%), compared to respondents from households without a child with a disability (68.2%). There was no difference in water source by household type; however, households with a child with a disability were less likely to have soap in the house. The education level of the respondents from the household with a child with a disability was low with 49.1% achieving primary school or less and 25.9% attending some secondary school (data not shown). Education level was not collected for households without a child with a disability. There was no difference in the proportion of respondents by the community; although a higher proportion of all respondents were living in the Chawama community. See Table 1.

The average age of the child with a disability was 10.8 years. Approximately one-third of the respondents from households with a child with a disability indicated their child had nutrition or feeding difficulties. The most common difficulty was that the child ate only a few foods (16.1%) and was unable to sit upright for feeding (15.4%). Other common difficulties included the child needing support eating, coughing and choking during feeding, and being unable to chew resulting in prolonged feeding times (data not shown).

Figure 2 presents data on HDDS and the target values for HDDS and IDDS using the average scores of the 33% of households with the highest diversity [29]. There was a significant difference in the household dietary diversity scale between the households. Households with a child with a disability had a score of 4.8 vs. 6.1 for other households (range 0–12). The IDDS scores were 3.5 vs. 4.4, respectively (range 0–8). The average household dietary diversity for both populations was below the target HDDS score of 8.0 and IDDS score of 5.6.

Based on individual items, families with a child with a disability were more likely to have eaten nshima and vegetables, compared to households without a child with a disability. All other food categories were more common in households without a child with a disability. See Table 2.

An unadjusted linear regression of participants with or without a child with a disability and the HDDS showed that households with a child with a disability had significantly lower scores of the HDDS, with an average difference of 1.34 points less, compared to households without a child with a disability (coef = −1.34, 95% CI: −1.58:−1.10). Adjusting for age, sex, community, the extent COVID-19 impacted access to food and presence of soap in the house attenuated the association; however, the HDDS score has an average difference of 1.08 points less in dietary diversity among households with a child with a disability (coef = −1.08, 95% CI: −1.33:−0.84). Similarly for the IDDS, models showed an average difference of nearly one point less in the IDDS for households with a child with a disability in unadjusted (coef = 0.97, 95% CI: −1.12:−0.817) and slightly attenuated, adjusted models (coef = −0.79, 95% CI: −0.95:−0.63). See Table 3. Therefore, for both HDDS and IDDS, there was an approximate difference of an average of one food category (0.79–1.08 adjusted) less consumed the previous day among households with a child with a disability, compared to households without.

## 4. Discussion

The need for nutritional and food support for families of children with disabilities have existed long before the COVID-19 pandemic. Containment measures, such as shelter-in-place orders reduced access to health services, food security and school closures highlighting these pre-existing vulnerabilities. Results revealed that households with a child with a disability had lower household and child-level dietary diversity, compared to households without a child with a disability. Additionally, households with a child with a disability were more likely to indicate that COVID-19 had disrupted their dietary diversity to a great extent. These findings uncover the complexity of malnutrition and disability, particularly in LMICs. To our knowledge, there are no published articles on this topic.

Many factors associated with disability are linked to malnutrition, including poverty and ill health [30]. One-third of the households with a child with a disability reported feeding and nutrition challenges that further impair their ability to ingest a wide range of food groups safely. Consumption of a higher number of food groups is associated with childhood growth and access to macro and micronutrients necessary for optimal development [31]. In this sample, while “nshima” and “vegetables” were significantly consumed more in the previous 24 h in households with a child with a disability, fruit, iron-, and protein-rich foods were less available, similar to patterns observed in other studies from Zambia [2]. These compounding factors make achieving dietary diversity increasingly difficult for children with a disability and put them at even further risk for secondary disabilities [32]. It is not clear why households with children with disabilities reported a higher vegetable intake. Nshima is commonly served with a tomato and onion relish [33] which are both low cost in the compound and easily served to a child with a disability. In this sample, dietary diversity was low regardless of feeding challenges, suggesting general levels of high poverty. The households without children with disabilities also have less than optimal dietary diversity, reflecting similar economic challenges, yet it is still more diverse than those with disabilities.

Global health fields of malnutrition and disability both address problems affecting vulnerable populations, but have often worked separately from one another. Good nutrition and dietary diversity are often discussed in relation to preventing disability and disability is often seen as a specialist field and may not be included in nutrition or childhood development education [34]. Yet, it is critical that these development programs also recognize causality among individuals born with or who have acquired a disability that presents oral motor/mechanical difficulties leading to a decreased nutrient intake [8], as well as experiences of stigma which leave the individual with disabilities and their family on the fringes of society [18,32]. The reciprocal links between malnutrition and disability are major public health problems and require increased international attention, investment, and collaboration.

In order for children with disabilities to meet their full health potential, there is a need to upscale support for parents and caretakers. Nutrition and food access are socio-economic issues requiring multi-sectoral investment into education, nutrition support and health care. The cultural and social causal pathways leading to increased malnutrition among families with a child with a disability requires recognition from the global health community followed by action-oriented recommendations and strategies for local government and policy makers.

As governments, the private sector, civil society, development and humanitarian partners continue the work of improving food systems, developing nutrition programs, and implementing COVID-19 recovery programs, special consideration must be given to families with children with disabilities. The Social Cash Transfer program did not have enough funding to support all families; hence a referral by the CCGs did not necessarily result in financial support [14]. Improving dietary diversity and preventing malnutrition among children with disabilities will require committed and more inclusive public health systems and focused government interventions [35]. Further investigation and community led research is needed to develop effective implementation strategies with practical solutions. This is particularly true given the fact that children with disabilities are often hidden in society; therefore, intentional and focused early intervention programs should be tested to assure adequate reach.

Limitations: This study has limitations. First, participants of Kusamala+ have been receiving social support and referrals through the intervention, thus their dietary diversity may have been higher than had they not been engaged in the program. Despite referrals to government support, their dietary diversity remained low. As noted, data was collected in September 2021, which was a challenging time, thus this snapshot may represent the worst dietary diversity rather than typical. The tool was designed to reflect all food consumed by household members; however, it is possible it did not include all foods available to everyone. Collecting data at the endline only excludes those who were lost to follow-up, which may influence the average characteristics of the households. A rapid health assessment was conducted in 2020 noting nutritional deficits [14], which is why the HDDS was added to the data collection at the endline. Finally, the HDDS and IDDS are only a snapshot in time and may not fully include all foods unknowingly eaten by all household members. It also does not provide information on dietary intake differences within the household.

## 5. Conclusions

Families with a child with a disability face many challenges in reaching their full health potential. The risk for food insecurity was high even before the COVID-19 pandemic, but this global crisis has amplified the barriers many of these families face in achieving dietary diversity for their households. These challenges are key human rights concerns, including the right to food. Families with a child with a disability cannot survive or thrive without external supports. Nutrition and disability are intricately connected and must be addressed together. Government programming must implement focused programs for children with disabilities and prioritize their households for financial and food supplementation. The Zambian government, in 2021, was motivated to shift funding to community driven initiatives. This research strives to generate evidence to inform inclusive program planning and responsible policy decisions aimed at improving health outcomes for children with disabilities and their families.

## Figures and Tables

**Figure 1 ijerph-20-02343-f001:**
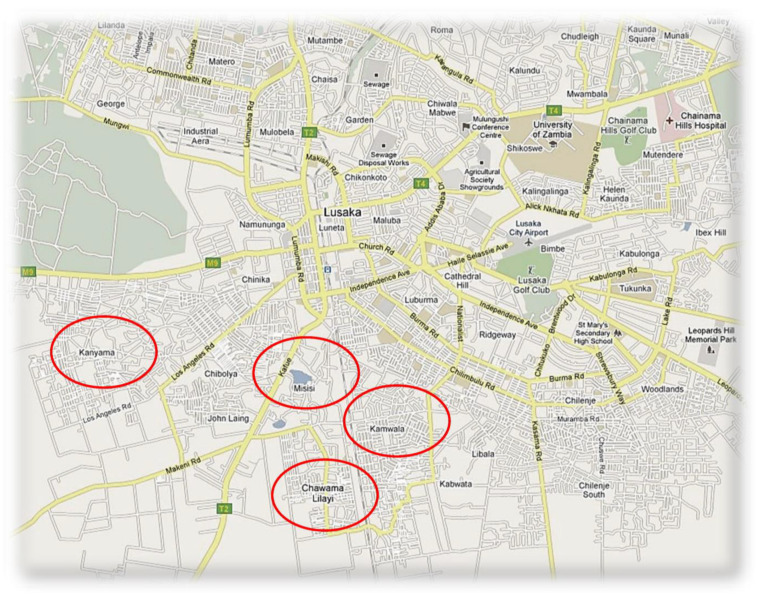
Map of Lusaka, Zambia with the four compounds indicated.

**Figure 2 ijerph-20-02343-f002:**
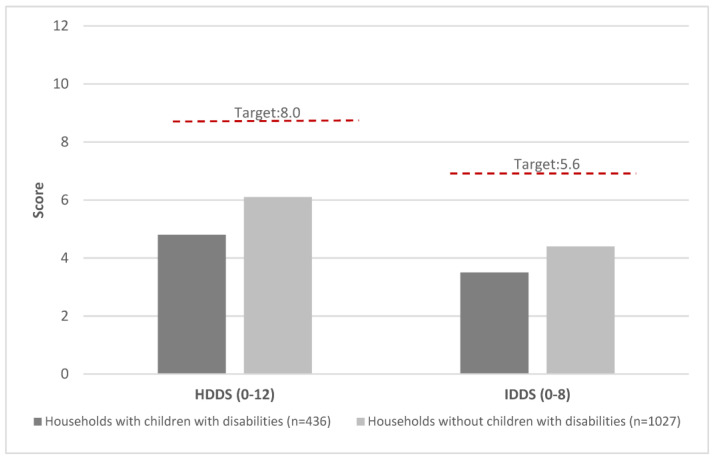
Household dietary diversity score and individual dietary diversity score among households with children with disabilities versus those without, Lusaka Zambia 2021.

**Table 1 ijerph-20-02343-t001:** Demographic descriptions of households with a child with disabilities and households without a child with a disability, Lusaka, Zambia 2021.

Variable	Household with a Child with a Disability *n* = 436 (30%)	Household without a Child with a Disability *n* = 1027 (70%)
Adult Age (mean (SD))	39.3 (0.54) *	37.9 (0.38)
Female *n* (%)	416 (95.3) **	700 (68.2)
Child age (mean (SD))	10.8 (5.4)	
Community *n* (%)		
Kanyama	116 (26.6)	234 (22.8)
Kanyama West	52 (11.9)	115 (11.2)
Misisi	88 (20.2)	243 (23.7)
Chawama	180 (41.3)	434 (42.4)
Access to water from a community borehole *n* (%)	218 (50)	472 (46)
We have access to soap at home *n* (%)	344 (79) **	924 (90)
My child has difficulties with feeding *n* (%)	128 (29.4)	
Extent to which COVID-19 impacted ability to obtain food *n* (%)		
To a great extent	139 (31.8) **	226 (22.0)
To a small extent	167 (38.3)	510 (49.7)
It has not changed	126 (29.0)	275 (6.9)
It has improved	4 (0.9)	15 (1.5)

Note: * *p*-value < 0.05; ** *p*-value < 0.01.

**Table 2 ijerph-20-02343-t002:** The percentage of participants who reported consumption of each food group using the household dietary diversity score in households with and without a child with a disability, Lusaka, Zambia 2021.

	Household with a Child with a Disability (*n* = 436)	Household without a Child with a Disability (*n* = 1027)
Item specific	*n* (%)	*n* (%)
Nshima	432 (99.1) *	990 (96.4)
Potato	80 (18.4) *	305 (29.7)
Vegetables	408 (93.6) *	893 (87.0)
Fruit	49 (11.3) *	338 (32.9)
Meat	121 (27.8) *	471 (45.9)
Eggs	102 (23.4) *	347 (33.8)
Fish	94 (21.6) *	303 (29.8)
Beans	115 (26.4) *	398 (38.8)
Dairy	48 (11.0) *	299 (29.1)
Fats	172 (39.5) *	562 (54.7)
Sweets	245 (56.2) *	716 (69.7)
Condiments	220 (50.5) *	676 (65.8)

Note: * *p* < 0.01.

**Table 3 ijerph-20-02343-t003:** Unadjusted and adjusted linear regression models comparing dietary diversity scores between families with and without a child with a disability, Lusaka Zambia 2021.

		Unadjusted		Adjusted	
		Coef	95% Confidence Interval	Coef	95% Confidence Interval
IDDS	Respondent with or without a child with a disability	−0.97	−1.12:−0.817	−0.79	−0.95:−0.63
	Age			−0.01	−0.01:0.00
	Sex			−0.27	−0.44:−0.09
	Compound			−0.08	−0.14:−0.02
	Extent COVID-19 impact food—Great extent			0.00	
	Small change			0.42	0.24:0.60
	It has not changed			0.43	0.24:0.63
	It has improved			1.00	0.37:1.64
	Soap available			−0.50	−0.71:−0.28
HDDS	Respondent with or without a child with a disability	−1.34	−1.58:−1.10	−1.08	−1.33:−0.84
	Age			−0.01	−0.02:−0.0002
	Sex			−0.23	−0.49:0.34
	Compound			−0.15	−0.24:-0.07
	Extent COVID-19 impact food—Great extent			0	
	Small change			0.62	0.35:0.89
	It has not changed			0.70	0.41−1.00
	It has improved			1.64	0.69:2.56
	Soap available			−1.04	−1.37:−0.72

## Data Availability

De-identified data available upon request from the corresponding author.

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
