# Peer review of "Household Dietary Diversity among Households with and without Children with Disabilities in Three Low-Income Communities in Lusaka, Zambia"

_ijerph, 2023, doi:10.3390/ijerph20032343_

Round 1

Reviewer 1 Report

The authors assess the dietary diversity of low-income households in three compounds in Lusaka, Zambia. They put a particular focus on the difference between households with children with disabilities and households without a child with a disability. They find that households with a disabled child had significantly lower HDDS and IDDS. They conclude that there is a need to strengthen support for these more vulnerable households.

The issue is relevant and warrants investigation. The results are consistent with expectations. However, I think there may be potential problems with the analysis. The authors argue that they analyze “comparable households with and without a child with a disability”. This is a matter of concern to me, since I am not fully convinced this is true. Apart from this, there are other potential problems, which I point out below. Anyway, I believe that the HDDS and IDDS results could be valid and deserve attention by themselves, and I think the paper could be improved by addressing thoroughly the following comments and suggestions:

·       Is the sample of households with children with disabilities and households without a child with a disability really comparable except for the fact that they have a disabled child? This is not necessarily the case. Besides, Table 1 shows both populations are in principle not comparable. I would suggest finding evidence on whether the samples are comparable.

·       I am also concerned about the fact that interviewed households with children with disabilities belong to the endline survey for a specific program. Is there any possibility of selection bias arising from potential systematic characteristics of participants in the program apart from the fact that households have disabled children?

·       The authors have acknowledged as a limitation the fact that “participants of Kusamala+ have been receiving social support and referrals through the intervention, thus their dietary diversity may have been higher than had they not been engaged in the program” (lines 258-260). I think the limitation is serious enough to be addressed from the beginning of the article and deserves further discussion.

·    HDDS and IDDS need to be explained better. The food categories of the HDDS and IDDS for children do not coincide with the usual ones. I am aware the questions regarding the calculation of the HDDS can be expanded to introduce any locally available foods within a group, —in this case nshima— to adapt them to the context of the study. However, I thought the 12 original food groups had to be reported later, when constructing the HDDS. If this is not a typo, I would suggest documenting in detail how the authors have constructed the index.

·      In addition, in parts of the article only 11 categories have been listed for the HDDS.

·     Regarding the HDDS and the IDDS, there is no discussion of the implications and potential problems/weaknesses of using this variable.

·      In addition, the authors describe the variable as “a proxy measure of these described causes and consequences of household economic ability to access a variety of foods and is an indicator of food security in general” (lines 48-51). I am not sure it is correct to say it is “a proxy measure of these described causes and consequences”, even if there is a potential causal relationship between them.

·    The authors state that households were asked “the HDDS”. I do not think this is correct, is it? They should be asked whether they have consumed specific foods or food groups, shouldn’t they?

·      I would like to know whether the authors are calculating the IDDS of each household member. It is not clear from the text. The IDDS of which exact household member is then used for the analysis?

·       The analysis section needs to be further explained. It lacks enough detail to be fully understood.

·    I would suggest avoiding naming households without a disabled child as “community members”. After all, households with disabled children are also part of the community.

·       The presentation of results needs to be improved.

·       Tables need to be improved too.

·    I would suggest showing in a table the complete results on the common difficulties for disabled children concerning their feeding. I think it is interesting and useful for understanding the issue.

·       I find the descriptive data rather shallow. Would it be possible to expand them a bit?

·      Are the authors assessing the impact of COVID-19 on dietary diversity or is it just an event that might have aggravated an existing problem? It is a bit confusing how this matter is dealt with throughout the paper. I would suggest making this clear throughout the paper.

Author Response

Please see my responses to reviewer comments in attached document. 

Reviewer 2 Report

The manuscript concerns the need to support families of the child with a disability and low-income families in Zambia as a governmental problem not only in this country but as a global problem.

The subject is not new, but the circumstances of COVID-19 increased the importance of quick practical solutions.

I believe missing in the discussion paragraph is comparing results with other research results, if there are any.

 Moreover, the text needs  English checking.

Suggestions for some minor corrections are below.

In abstract

 Line 21: for community house-20 holds. (lack of dot?) Household with a child

In Introduction

Line 45: Consumption of diets that are not diversified have been linked- or has been linked

Line 61: Data was in collected- data were collected?

Line 67: living in three low-resource compounds in Lusaka, Zambia. Data in the table show four areas. Please, show it on the map.

Materials and methods

Line 75: what does mean word endline?

Results

Table 3: Extent COVID-19 impact food- impact on food?

Discussion

Line 215: to safely ingest a wide range of food groups

or? to ingest a wide range of food groups safely.

Line: vegetables or vegetables?

Conclusions

Line 275 for child or for children

References;

1 and  22: one time The Journal of Nutrition, the other J. Nutr,

Author Response

My responses to reviewers are attached.

Round 2

Reviewer 1 Report

·       The authors point out in their cover letter the population in the three communities is quite homogeneous. Are they suggesting that this alone could be reason enough to consider the sample of households with children with disabilities and households without a child with a disability comparable except for the fact that they have a disabled child? Even if I am willing to accept the reasoning, I am still worried about the fact that Table 1 shows both populations are in principle not comparable. I would still suggest finding evidence on whether the samples are comparable, explaining more clearly why the reader should consider both samples comparable and making an additional effort to explain why Table 1 shows significantly different characteristics between the two populations.

·       I am still concerned about the fact that interviewed households with children with disabilities belong to the endline survey for a specific program. The authors acknowledge that there is a possibility of selection bias arising from potential systematic characteristics of participants in the program apart from the fact that households have disabled children. This point needs to be better justified (not only in the cover letter, but in the article as well).

·       Regarding the explanation on how HDDS was constructed, the part on the inclusion of locally available foods within a group — in this case nshima— needs to be more clearly explained.

·       How can the Individual Dietary Diversity Scores (IDDS) be calculated if individual dietary diversity was not collected?

·       Is pairwise correlation only used to examine “the relationship between nshima and vegetable consumption as culturally a tomato sauce is typically served with nshima” (lines 168-170)? The analysis section still needs to be further explained. It lacks enough detail to be fully understood.

·       Households without a disabled child are still named sometimes “community members”. For example, in line 19 or in line 183.

·       Table 1: Why is the variable education level not shown? In addition, I think the results could be further elaborated.

·       Style and syntax need to be improved.

Author Response

Please see attached item by item response.
